# In Vitro Anti-NTHi Activity of Haemophilin-Producing Strains of *Haemophilus haemolyticus*

**DOI:** 10.3390/pathogens9040243

**Published:** 2020-03-25

**Authors:** Brianna Atto, Roger Latham, Dale Kunde, David A Gell, Stephen Tristram

**Affiliations:** 1School of Health Sciences, University of Tasmania, Newnham Drive, Launceston, TAS 7248, Australia; dale.kunde@utas.edu.au; 2School of Medicine, University of Tasmania, 17 Liverpool Street, Hobart, TAS 7000, Australia; roger.latham@utas.edu.au (R.L.); david.gell@utas.edu.au (D.A.G.)

**Keywords:** Haemophilus influenzae, Haemophilus haemolyticus, respiratory probiotic, respiratory infection, heme, heme-binding protein, otitis media, chronic obstructive pulmonary disease

## Abstract

Nontypeable *Haemophilus influenzae* (NTHi) is a leading causative organism of opportunistic respiratory tract infections. However, there are currently no effective vaccination strategies, and existing treatments are compromised by antibiotic resistance. We previously characterized *Haemophilus haemolyticus* (Hh) strains capable of producing haemophilin (HPL), a heme-binding protein that restricts NTHi growth by limiting its access to an essential growth factor, heme. Thus, these strains may have utility as a probiotic therapy against NTHi infection by limiting colonization, migration and subsequent infection in susceptible individuals. Here, we assess the preliminary feasibility of this approach by direct in vitro competition assays between NTHi and Hh strains with varying capacity to produce HPL. Subsequent changes in NTHi growth rate and fitness, in conjunction with *HPL* expression analysis, were employed to assess the NTHi-inhibitory capacity of Hh strains. HPL-producing strains of Hh not only outcompeted NTHi during short-term and extended co-culture, but also demonstrated a growth advantage compared with Hh strains unable to produce the protein. Additionally, *HPL* expression levels during competition correlated with the NTHi-inhibitory phenotype. HPL-producing strains of Hh demonstrate significant probiotic potential against NTHi colonization in the upper respiratory tract, however, further investigations are warranted to demonstrate a range of other characteristics that would support the eventual development of a probiotic.

## 1. Introduction

The bacterium nontypeable *Haemophilus influenzae* (NTHi) is commonly associated with upper respiratory tract (URT) colonization in healthy adults [1]. However, migration to other sites in the respiratory tract frequently occurs in children, the elderly and individuals with underlying respiratory diseases; making NTHi a leading cause of mucosal infections [2]. In particular, enormous global morbidity is attributed to otitis media and exacerbations of chronic obstructive pulmonary disease, which are accompanied by long-term health complications and considerable mortality, respectively [3,4]. NTHi has also gained attention as an increasingly important cause of invasive infections [5,6].

There are currently no effective vaccination strategies for the prevention of NTHi infections, and treatment has been complicated by the rapid development of antibiotic resistance to first- and second-line antibiotics. Resistance is predominantly mediated by β-lactamase production [7]; however, the emergence and spread of β-lactamase-negative, ampicillin-resistant strains in many regions of the world is of substantial concern with treatment failure also being reported in response to macrolides [8,9,10] and fluoroquinolones [11,12,13]. 

NTHi infection is preceded by successful colonization of the URT, and survival in this environment relies on the bacterium’s ability to acquire the vital growth factor, heme [14]. There is also evidence to suggest heme acquisition genes are important modulators of NTHi virulence factors [15], demonstrated by the increased prevalence in disease-causing strains from the middle ear, compared with colonizing throat strains [14]. Deletion of multiple genes related to heme-iron scavenging, utilization and regulation has been shown to significantly reduce NTHi virulence, disease severity and duration in animal models of otitis media [16,17]. Similarly, an isogenic mutant of two heme acquisition pathways was unable to sustain bacteraemia or produce meningitis in a rat model of invasive disease [18]. Thus, heme acquisition pathways represent potentially high value targets for the development of novel therapies for the eradication of NTHi from the respiratory tract [19,20].

NTHi is particularly susceptible to heme restriction as it lacks the necessary enzymes for its synthesis and relies solely on scavenging heme from the host, either in the form of free heme or bound to host carrier molecules [16,21,22,23]. Evidence from our laboratory suggests that closely related commensals may present a competitive challenge for heme acquisition in the URT. Previously, we discovered *Haemophilus haemolyticus* (Hh) strains that exhibited inhibitory activity against NTHi [24,25]. Further investigation revealed this inhibition was mediated by the production of a heme-binding protein, haemophilin (HPL), that restricted NTHi growth by limiting its access to heme [25]. Thus, these strains may have utility as a probiotic therapy against NTHi infection by limiting colonization, migration and subsequent infection in susceptible individuals. Hh strains with anti-NTHi properties have other characteristics that support their potential use as probiotics. Firstly, they share the same upper respiratory niche as NTHi [1] and more importantly, although they have occasionally been reported as pathogens of sterile sites in immunocompromised patients [26], there is convincing evidence that they are not opportunistic pathogens of the respiratory tract [27,28,29].

Here, we aim to determine the potential of a future probiotic approach by assessing in vitro competition between NTHi and Hh strains with varying capacity to produce HPL. 

## 2. Results and Discussion

### 2.1. Validation of a Triplex Real-Time PCR for Quantification of NTHi, Hh and Detection of HPL

The *HPL* amplicon was confirmed to be specific and sensitive for the detection of the five previously identified *HPL* sequence variants [25] by in silico investigations and by PCR. Specificity of the *HypD* and *SiatT* targets was also confirmed by PCR. Complete results of PCR assay validation are detailed in Appendix A. The low limit of quantification values for the *HypD* and *SiaT* assays in triplex format were 2 × 10^−5^ ng and 2 × 10^−4^ ng, corresponding to 10 and 100 GE, respectively. The lower limit of detection for the *HPL* assay was 10 GE (Appendix A). The upper limits of detection/quantification were not explicitly determined as expected DNA levels from sample were unlikely to exceed the maximum 2 ng tested. 

Given the high volume of samples generated from growth experiments, a cheap and high-throughput gDNA extraction method was required to reliably distinguish and quantify NTHi and Hh in co-culture. Extraction utilizing thermal lysis has previously been shown to be an efficient and cost-effective method to harvest bacterial gDNA for quantitative real-time PCR from suspensions of several bacterial species in a range of sample matrices [30,31,32,33,34,35]. Crude DNA extraction methods are also prone to contamination with PCR inhibitors originating from sample matrices [33,36]. There are also reports of intra- and inter-species differences in DNA extractions efficiencies [33,36,37]. PCR quantification of gDNA extracted by thermal lysis was validated and found to be comparable to quantification by OD_600_ and colony counts (Appendix A). Complete results of thermal extraction validation, including detection of PCR inhibitors (Appendix A) and extraction efficiency Appendix A. 

### 2.2. Baseline NTHi-Inhibitory Activity of Hh Strains Containing the HPL ORF

We previously identified clinical isolates of Hh with several different *HPL* open reading frame (ORF) sequences and variable inhibitory activities even between strains containing identical *HPL* ORF sequences (Appendix A) [25]. In order to determine the basis of this phenotypic variation and predict inhibitory potential, selection of Hh strains (Hh-RHH122, Hh-NF4 and Hh-NF5) for investigation was based on identical sequence similarity to the Hh-BW1 *HPL* ORF, previously identified as having the highest NTHi-inhibitory activity [25]. Based on results from the well diffusion assay, isolates Hh-BW1, Hh-RHH122 and Hh-NF5 were categorised as having the Hh-HPL^+^ phenotype; no inhibitory activity was detected from Hh-NF4 broth supernatants, categorising it as Hh-HPL^−^. Hh strains that did not possess the *HPL* ORF (Hh ATCC 33390 and Hh-BW1*^HPL^*^-KO^) were confirmed to be Hh-HPL^−^. The degree of inhibitory activity varied between the Hh-HPL^+^ isolates and was comparably highest in Hh-BW1 and Hh-RHH122, approximately twice the activity measured for Hh-NF5 (Appendix A). 

### 2.3. HPL Expression Correlates with the Hh-HPL^+^ Phenotype

Given that *HPL* ORF sequence identity was not predictive of NTHi-inhibitory capacity, baseline expression of *HPL* was investigated. The *hypD* target was validated as the housekeeper gene (Appendix A), and the optimal growth phase for *HPL* expression analysis was determined (Appendix A). Baseline expression of *HPL* was highest in Hh-BW1 and Hh-RHH122, significantly lower in Hh-NF5 (*p* < 0.0001), and completely absent in Hh-NF4 (Figure 1). Expression patterns correlate with the NTHi-inhibitory capacity of Hh strains, suggesting a connection between expression of *HPL* and the NTHI-inhibitory phenotype resulting from production of the HPL protein.

### 2.4. The Hh-HPL^+^ Phenotype Confers a Competitive Advantage against NTHi 

A co-culture assay was used to test the ability of Hh with different levels of HPL production to compete with NTHi. The growth rate of NTHi was significantly impaired during competition with all Hh-HPL^+^ strains, compared with growth without competition (*p* < 0.0001) (Figure 2A). This inhibitory effect was more pronounced during competition with strains Hh-BW1 and Hh-RHH122, compared with Hh-NF5, replicating inhibitory patterns observed in the well diffusion assay. The growth rate of NTHi during competition with Hh-HPL^−^ was not significantly affected (Figure 2A), suggesting that the inhibitory effect observed was a unique characteristic of Hh-HPL^+^ strains. Loss of the Hh-HPL^+^ phenotype in Hh-BW1*^HPL^*^-KO^, compared with the wild-type Hh-BW1, is evidence for a causative effect of the *HPL* gene on competition. Correlation between competition outcomes and *HPL* gene expression in Hh-BW1, Hh-RH122, Hh-NF5 and Hh-NF4 strains is also consistent with a hypothesis that strains with higher *HPL* expression compete with NTHi more effectively.

For commensals and pathogens living in or invading human tissues, iron-containing heme is often a limiting nutrient, particularly in the respiratory tract where concentrations are considered to be low [38]. This is particularly true for heme auxotrophs including NTHi and Hh; for these species’ survival in the URT niche is dependent on their ability to outcompete host proteins and co-existing bacterial populations for heme [16]. We previously demonstrated that the NTHi-inhibitory mechanism of HPL is associated with it’s ability to bind heme in a form inaccessible to NTHi and that inhibitory activity is lost in conditions where heme concentration exceeds the binding capacity of HPL [25]. While levels of heme/iron are considered to be low in the respiratory tract, there is indirect evidence for increased levels in airways of smokers, chronic obstructive pulmonary disease and cystic fibrosis, which may contribute to increased susceptibility to infection in these individuals [38]. Thus, it was important to assess the effectiveness of HPL with varying concentrations of heme to ensure inhibitory effect in a range of in vitro conditions reflecting possible in vivo scenarios. The NTHi-inhibitory capacity of HPL was maintained even in conditions of high heme availability (15 µg mL^−1^), albeit to a lesser degree than lower heme concentrations (0.0–3.8 µg mL^−1^) (Figure 2A). This suggests that levels of HPL produced by Hh are sufficient to limit NTHi’s access to heme in a dynamic in vitro system, even under excess heme conditions unlikely to be encountered in vivo [38]. 

### 2.5. The Hh-HPL^+^ Phenotype is Associated with a Growth Advantage 

Interestingly, all Hh-HPL^+^ strains exhibited a pattern of enhanced growth in response to NTHi competition (*p* < 0.0001) (Figure 2C). This effect was maintained across all heme concentrations and was more pronounced for Hh-BW1 and Hh-RHH122. The converse was observed in Hh-HPL^−^ strains that appeared to experience poorer growth in response to competition with NTHi (Figure 2D). This indicates that NTHi is able to outcompete Hh only in the absence of the Hh-HPL^+^ phenotype, which may be a reflection of the highly efficient set of heme-scavenging systems possessed by NTHi. 

Given the correlation between competition outcomes and *HPL* gene expression, expression analysis was performed on Hh during competition with NTHi, relative to growth without competition. Upregulation of *HPL* was observed in all Hh-HPL^+^ in response to competition with NTHi, an effect that was more pronounced in Hh-BW1 and Hh-RHH122 (Figure 3). This may explain the enhanced growth rate of Hh-HPL^+^ strains in response to NTHi during the short-term competition assays (Figure 2C). 

These results show that expression of *HPL* has a significant impact on the NTHi-inhibitory capacity of Hh-HPL^+^ strains and eventual therapeutic utility in an in vivo setting. Therefore, the huge differential expression of *HPL* amongst Hh-HPL^+^ strains must be considered in the future when selecting a potential probiotic candidate for further evaluation. However, our understanding of HPL regulation is still rudimentary. Further investigation into potential upstream regulatory components or post-translational modification is needed to elucidate the inter-strain differences in HPL production and/or biological activity despite complete ORF sequence identity. 

### 2.6. NTHi Fitness Dramatically Decreases during Extended Co-Culture with Hh-HPL^+^

Short-term competition may highlight the potency of HPL-mediated inhibition but is not representative of in vivo competition dynamics. Thus, a longer-term study was employed to assess the competition between NTHi and Hh-HPL^+^ over a period of 6 days (12 generations). The competitive advantage of Hh-HPL^+^ strains was evident within the 2nd generation (24 h) for Hh-BW1 and Hh-RHH122 and 4th generation (48 h) for Hh-NF5 (Figure 4A). Speculatively, the stunted inhibitory activity exhibited by Hh-NF5 may be due to the lower *HPL* expression levels exhibited by this strain, resulting in slower accumulation of HPL over the course of the assay. The fitness of NTHi over subsequent generations decreased significantly until complete loss of fitness during the final generations. Competition with Hh-HPL^−^ strains did not significantly affect the overall fitness of any of the NTHi strains over the 6 day period. However, a small decrease in NTHi fitness was observed after 24 h, followed by complete recovery (Figure 4A). This may have arisen from competition for heme prior to the onset of maximum HPL production.

To show that loss of fitness of NTHi was not unique to NTHi strain ATCC 49247, additional reference strains NCTC 11315 and ATCC 49766 were tested in competition with Hh-BW1. All three NTHi strains responded in the same manner, culminating in a total loss of NTHi fitness at the end of the 6 day period (Figure 4B).

## 3. Materials and Methods

### 3.1. Bacterial Growth Conditions 

#### 3.1.1. Bacterial Strains 

Hh strains used in this study (BW1, RHH122, NF1, NF4 and NF5) have previously been isolated and screened for the *HPL* open reading frame (ORF) [24,25]. An *HPL* knockout (BW1*^HPL^*^-KO^) of the model HPL-producing strain of Hh (Hh-BW1), constructed using insertional inactivation as previously described [25], and Hh ATCC 33390 (PCR negative for the *HPL* ORF) were included as noninhibitory controls. NTHi strains were ATCC 49247, ATCC 49766 and NCTC 11315.

NTHi and Hh isolates were propagated from liquid nitrogen frozen glycerol stock, followed by two overnight passages on chocolate agar (CA) at 37 °C with 5–10% CO_2_ prior to experimentation. Strains were grown in supplemented Tryptone Soy Broth (sTSB), which consisted of tryptone soy broth (TSB) (Oxoid Ltd., Basingstoke, UK) supplemented with 2% (v/v) Vitox^®^ (Oxoid Ltd.) and 15 µg mL^−1^ of porcine hematin (ferriprotoporphyrin IX hydroxide, Sigma-Aldrich). Exposure to nongrowth conditions was minimized by maintaining suspensions and diluents at 37 °C.

#### 3.1.2. Propagation of Heme-Replete Populations for Growth Experiments

Strains were propagated under heme-replete conditions prior to use in competition experiments to replenish bacterial heme stores and minimise external stressors that may influence the outcome of competitive growth [25,39,40]. Bacterial suspensions of ~1.0 OD_600_ were made in TSB from 8–10 h growth on CA and diluted 1:10 in pre-warmed sTSB (5 mL). Broths were incubated for 12 h at 37 °C aerobically with shaking (220 RPM), centrifuged at 4000 × g for 5 min at 37 °C and resuspended in fresh, pre-warmed TSB to an OD_600_ of 1.0 prior to use in growth experiments. 

### 3.2. Determination of NTHi-Inhibitory Activity

A well diffusion assay of broth supernatants was used to categorise Hh strains containing the *HPL* ORF as either inhibitory to NTHi (Hh-HPL^+^) or noninhibitory (Hh-HPL^–^), as previously described [24]. This assay was also used to establish the relative inhibitory activity of each strain. Testing was conducted on two indicator NTHi strains (ATCC 49427 and clinical isolate NTHi-L15). Media supernatants from strains negative for the *HPL* ORF (Hh ATCC 33390 and Hh-BW1*^HPL^*^– KO^) were included as controls.

### 3.3. Triplex Real-Time PCR for the Quantification of NTHi, Hh and Detection of HPL

A real-time quantitative triplex PCR assay was designed to quantify NTHi, Hh and detect the *HPL* ORF. The targets used for discrimination of Hh (*hypD*) and NTHi (*siaT*) have previously been described and validated [41]. For detection of the *HPL* ORF, primers were designed based on the *HPL* ORF of Hh-BW1 (GenBank MN720274) [25]. The FAM, HEX and TET channels were used for simultaneous fluorescence detection of *siaT*, *hypD* and *HPL*, respectively. Primer and probe sequences are detailed in Table 1. Primer specificity was confirmed by discontiguous megaBLAST analysis and PCR of a panel of *Haemophilus* spp. and multiple genera representing common URT flora. PCR assays were extensively optimised and evaluated for detection/quantification limits in triplex format. 

PCRs were performed using the CFX96 Touch^TM^ real-time PCR system (Bio-Rad) in 96-well optical plates. Polymerase activation was performed at 95 °C for 3 min, followed by 40 amplification cycles of denaturation at 95 °C for 15 s, and annealing at 62 °C for 1 min. Each reaction contained 0.25 μM of *hypD*, *siaT* and *HPL* primers, 0.1 μM LNA probes, 1× PrimeTime master mix (Integrated DNA Technologies) and 5 μL gDNA and molecular-grade water, to a total volume of 20 μL. Template gDNA was prepared by a thermal extraction protocol and tested in duplicate. Each run included a positive control for the *HPL* ORF (Hh-BW1), negative control (*H. parainfluenzae* ATCC 7901), no-template control and 10-fold dilutions of a standard containing 2 × 10^−8^ ng NTHi ATCC 49247 and Hh ATCC 33390 gDNA. Quantification of NTHi and Hh was expressed as genome equivalents (GE) calculated from the standard, as previously described [41]. Bacterial quantification from thermally extracted gDNA was validated against conventional bacterial quantification by optical density and colony counts. 

Complete details of PCR primer design, assay optimisation and gDNA extraction protocol evaluation are available in Appendix A.

### 3.4. Competition Assays 

#### 3.4.1. Short-Term Broth Competition 

Culture mixes were prepared by adding 100 µL of heme-replete preparations of a single Hh strain (Hh-BW1, Hh-BW1*^HPL-^*^KO^, Hh-RHH122, Hh-NF4, Hh-NF5 or ATCC33390) together with 100 µL of NTHi ATCC 49247 to 5 mL pre-warmed sTSB containing 0.0, 0.9, 3.8 or 15.0 ug mL^−1^ porcine hematin. Broths containing single strains were also prepared in parallel to determine baseline growth. Broths were incubated aerobically on an incubator shaker at 37 °C (220 RPM) for 16 h. At different time intervals, aliquots of 50 µL were taken for thermal gDNA extraction and subsequent triplex PCR quantification of GE. Aliquots of 500 µL were taken at 8 h for quantification of *HPL* expression. Purity of broth growth was checked by plating on CA after 16 h incubation. 

Statistical comparisons were made between strains grown with a competitor and baseline growth by calculating the change in the number of cells per hour (growth rate) using the following formula:$$ln\frac{N_{t}}{N_{0}}=\alpha \left(t-t_{0}\right)$$
where *N_t_* is the number of cells (measured as GE) at time *t*, *N*_0_ is the number of cells at time zero (*t*_0_), and *α* is the growth rate where units are determined by the units of *t*. 

#### 3.4.2. Fitness Assay

Culture mixes were prepared by adding 100 µL of heme-replete preparations of each of the Hh strains and 100 µL of NTHi (ATCC 49247, ATCC 49766 or NCTC 11315) to 5 mL of pre-warmed sTSB containing 0.0, 0.9, 3.8 or 15.0 µg mL^−1^ porcine hematin. Broths were incubated aerobically on an incubator shaker at 37 °C (220 RPM) for 12 h prior to subculture (200 µL) in fresh sTSB (2 mL) containing the same concentration of heme as the inoculum. The process of 12 hourly incubation followed by subculture into fresh broth was repeated until 6 days had elapsed. After each 12 h incubation, aliquots of 50 µL were taken for boiled gDNA extraction and subsequent triplex PCR quantification of GE. Purity of broth growth was confirmed by plating on CA after each 12 h incubation. Fitness of NTHi at each time point was determined using the following equation [42]:$$w=\frac{ln\left(\frac{A_{t}}{A_{t_{0}}}\right)}{ln\left(\frac{B_{t}}{B_{t_{0}}}\right)}$$
where *w* is fitness, A and B are the population sizes of the two competitors, subscripts *t*_0_ and *t* indicate the initial and final time points in the assay. Growth after the first 12 h culture was used as baseline for fitness determination (*t*_0_). 

### 3.5. Expression Analysis 

#### 3.5.1. RNA Extraction, Purification and Quantification

Aliquots taken from broth growth were immediately added to two volumes of RNAprotect Bacteria Reagent (Qiagen) for immediate stabilization of bacterial RNA. Stabilized aliquots were normalized to an OD_600_ of 0.05 (approximately 5 × 10^7^ cells), pelleted by centrifugation for 10 min at 5000 × g and stored at −20 °C overnight. Bacterial lysates were prepared by resuspending pellets in 100 µL TE buffer (30mM Tris-Cl, 1 mM EDTA, pH 8.0) containing 15 mg mL^−1^ lysozyme and 20 µL proteinase K, vortexed and incubated at room temperature in an incubator shaker (1000 RPM) for 1 h. Following addition of 350 µL buffer RLT, samples were vortexed and centrifuged at 20000 × g for 2 min. The supernatant was purified following the manufacturers protocol for RNeasy Plus Mini Kit, which was semiautomated by the QIAcube (Qiagen). The iScript^TM^ cDNA Synthesis Kit (Bio-Rad) was used to produce cDNA for subsequent PCR. The validated triplex PCR was used to determine expression of *HPL* ORF in Hh strains, using *hypD* as the housekeeper gene.

#### 3.5.2. Expression Validation

Expression analysis was employed to determine baseline expression and suitability of prospective competitive test conditions for *HPL* expression. Given the kinetics of bacterial growth and the heme-binding capacity of HPL, time and heme availability were targeted as factors that may influence *HPL* expression. The *hypD* target was selected as a potential housekeeper gene and validated for test conditions. 

To validate expression analysis, heme-replete preparations of Hh-BW1 and the Hh-BW1*^HPL^*^-KO^ (100 µL) were added to 5 mL pre-warmed sTSB containing either 0.0 or 15.0 µg ml^−1^ of porcine hematin. Broths were incubated for 8 h, and aliquots of 500 µL were removed for RNA extraction and purification at 0, 4 and 8 h. Following validation, baseline *HPL* expression was quantified for isolates Hh-BW1, Hh-RHH122, Hh-NF4 and Hh-NF5 relative to Hh-BW1*^HPL^*^-KO^ from 8 h growth in sTSB without porcine hematin. 

### 3.6. Statistical Analysis 

Statistical analysis was performed using GraphPad Prism V7.04, 2017. Statistical significance was determined by comparison of growth data (growth rate or fitness) between strains grown with a competitor and baseline growth. Data were tested for normality using the Shapiro–Wilk test, followed by a two-way ANOVA with Dunnett’s multiple comparison test. Expression ratios and statistical significance were calculated with 2000 iterations by the Relative Expression Software Tool (REST; v 1.0, 2009) [43,44].

## 4. Conclusions

Previously, we identified an uncharacterized hemophore (designated HPL) produced by Hh which was able to inhibit NTHi growth by heme starvation [25]. Here, we aimed to further test the inhibitory capacity of Hh-HPL^+^ strains by direct in vitro competition with NTHi, for the purpose of determining their probiotic potential. These results provide a strong link between the NTHi-inhibitory phenotype, *HPL* expression and favourable outcomes during competitive growth with NTHi in vitro. Thus, Hh-HPL^+^ strains exhibit promising probiotic potential against NTHi colonization in the URT. Reduction or elimination of NTHi carriage from the URT proposes an effective means of limiting migration and subsequent infection in susceptible individuals. However, significant further investigation is required to determine if the inhibitory capacity demonstrated in this study extends to more complex models of NTHi colonisation and infection, such as cell culture systems and animal models. Further, studies investigating the safety profile of Hh-HPL+ strains and their ability to colonise the host are also required before the probiotic potential of such strains can be advanced.

## Figures and Tables

**Figure 1 pathogens-09-00243-f001:**
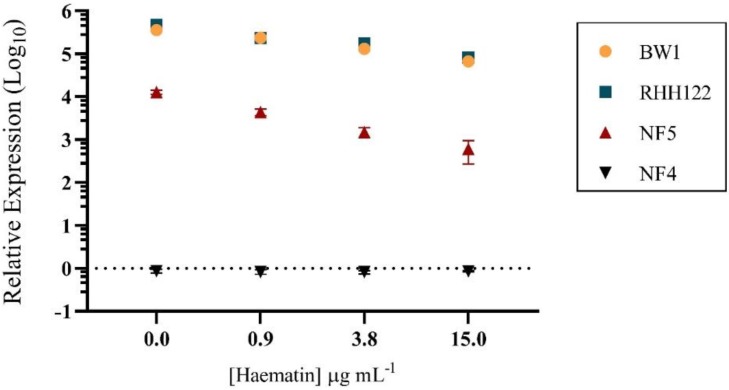
Baseline *HPL* expression. PCR-quantified expression of *HPL* for *Haemophilus haemolyticus* (Hh) strains containing identical *HPL* open reading frames (ORFs) (relative to Hh-BW1*^HPL^*^-KO^). Data points are represented as mean +/− SEM of four biological replicates, performed from duplicate RNA extractions.

**Figure 2 pathogens-09-00243-f002:**
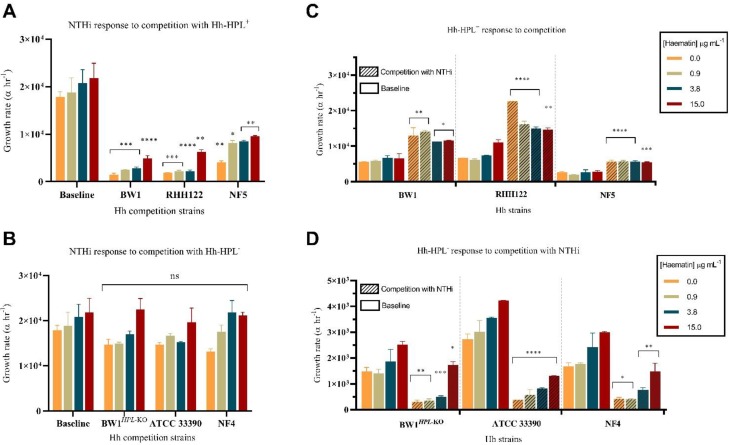
Short-term competition between *Haemophilus influenzae* (NTHi) and Hh. Calculated growth rates of NTHi in response to competition with (**A**) Hh-HPL^+^ or (**B**) Hh-HPL^−^. The growth rate for each (**C**) Hh-HPL^+^ and (**D**) Hh-HPL^−^ strain was also determined. Data points represented as mean +/− SEM of three separate experiments, performed in triplicate; *p* < 0.05 *, *p* < 0.005 **, *p* < 0.0005 ***, *p* < 0.0001 ****.

**Figure 3 pathogens-09-00243-f003:**
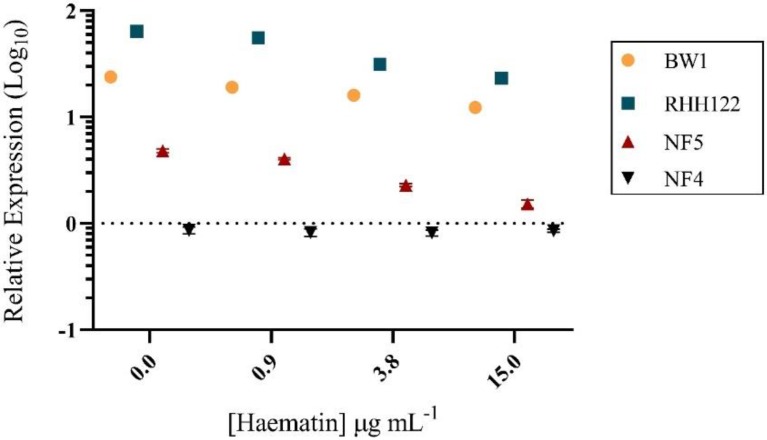
*HPL* expression during competition. PCR-quantified expression of *HPL* during competition with NTHi relative to individual growth. Data points are represented as mean +/− SEM of four biological replicates, performed from duplicate RNA extractions.

**Figure 4 pathogens-09-00243-f004:**
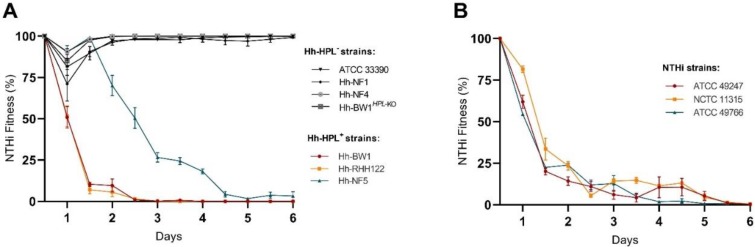
Fitness of NTHi strains during co-culture with Hh. Calculated fitness of NTHi in response to competition with Hh-HPL^+^ or Hh-HPL^−^ relative to growth of the competitor strain. (**A**) Competition between a single NTHi strain and multiple Hh, or (**B**) multiple NTHi against Hh-BW1. Data points represented as mean +/− SEM of three separate experiments, performed in quadruplicate.

**Table 1 pathogens-09-00243-t001:** Summary of primer and LNA probe sequences, and expected amplicon size for the *hypD*, *siaT* and *HPL* targets.

Primers and Probes	Sequence	Amplicon Size (bp)
**hypD Forward**	5′- GGCAATCAGATGGTTTACAACG	187
**hypD Reverse**	5′- CAGCTTAAAGYAAGYAGTGAATG	
**hypD LNA-probe**	/5HEX/CCA+C+AA+C+GA+G+AATTAG/3IABkFQ/	
**siaT Forward**	5′- AATGCGTGATGCTGGTTATGAC	138
**siaT Reverse**	5′- AATGCGTGATGCTGGTTATGAC	
**siaT LNA-probe**	/56-FAM/A+GA+A+GCAGC+A+G+TAATT/3IABkFQ/	
**HPL Forward**	5′- TATTCCTAATGATCCCGCT	120
**HPL Reverse**	5′ - TCTTTTTTCGCTACCCCT	
**HPL LNA-probe**	/5Cy5/AT+CCATTTA+TCGG+CACGTTCT/3IAbRQSp/	

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
