# Peer review of "In Vitro Anti-NTHi Activity of Haemophilin-Producing Strains of Haemophilus haemolyticus"

_pathogens, 2020, doi:10.3390/pathogens9040243_

Round 1

Reviewer 1 Report

The authors reported that several Haemophilus haemolyticus strains displayed the inhibitory effect to  non-typeable Haemophilus influenzae (NTHi) and suggested to serve as probiotics for this infectious disease. The authors also demonstrated that the inhibitory effects of these Haemophilus haemolyticus strains were associated with the production of haemophilin to limit the availability of heme for NTHi. The data were interesting but the main question is that there was no evidence to support the "probiotics" characteristics of these Haemophilus haemolyticus strains. Actually, Haemophilus haemolyticus has been reported to be associated with clinical diseases (J Clin Microbiol. 2012. 50(7): 2462–2465). The authors should clear demonstrate that these Haemophilus haemolyticus strains would not cause illness in animal test, as well as the lack expression of antibiotic resistant genes. Without such data, the authors could not claim that these Haemophilus haemolyticus strains are probiotics or to have probiotics potential.

Reviewer 2 Report

Atto et al investigated the  inhibitory potential of H haemolyticus  strains, especially of haemophilin compound. The topic is innovative and interesting. The chosen for the study methods were appropriate. The results were well presented and discussed.

comments:

Why have the authors  decided  to use for  designation  “ probiotic”? In my opinion, this should be avoided and it would be more appropriate to use “inhibitory potential”.  For probiotics, we need to be absolutely sure that they are not pathogenic, and as in the literature there are some reports for observation of H haemolyticus  in clinical samples/J Clin Microbiol. 2012 Jul; 50(7): 2462–2465; Journal of Antimicrobial Chemotherapy, Volume 71, Issue 1, January 2016, Pages 80–84 /, the authors should avoid using the word “probiotic”.

Why have the authors chosen to investigate specifically these five isolates of H haemolyticus – I understand that this is explained in the previous paper but please insert more information for them in this manuscript also

Line 266 –From results and discussion the readers understand that the competition assay was performed separately for each strain of H haemolyticus. In line 266/267 this is not clearly comprehensive.

Reviewer 3 Report

Although the manuscript provides interesting results in which HPL-producing strains of Hh demonstrate enormous probiotic potential against NTHi colonization in the upper respiratory tract, I have some minor and major concerns that need to be clarified and addressed before further consideration.

Minor concerns
1. The merits and limitations of the study should be highlighted and discussed adequately.
2. The safety profile of the probiotic potential of haemophilin-producing strains of Hh as well as the possibility of administration form should be clarified and discussed.
4. The future scope of the study should be also discussed.
Major Concerns
1. Some additional experiments are needed to characterize the probiotic properties. For instance, determination of the tolerance to low pH as well as the tolerance to bile salts.
2. The evaluation of adhesion capacity, hydrophobicity, and auto-Aggregation of haemophilin-producing strains of Hh should be performed to evaluate the probiotic properties.
In my opinion, the performed experiments are not enough to confirm the claim of the potential use of haemophilin-producing strains of Hh as a probiotic, where additional experiments are required to confirm such a claim.
I highly recommend the authors take into consideration the above-mentioned concerns during the revision.

Round 2

Reviewer 1 Report

The manuscript has been revised as the suggestion that to remove "probiotics" in the title and made some notes in the descriptions that it requires further characterizing the probiotics properties in the future.

Reviewer 3 Report

The manuscript has been significantly improved.